# Mosquito Salivary Proteins and Arbovirus Infection: From Viral Enhancers to Potential Targets for Vaccines

**DOI:** 10.3390/pathogens12030371

**Published:** 2023-02-23

**Authors:** Alejandro Marín-López, Hamidah Raduwan, Tse-Yu Chen, Sergio Utrilla-Trigo, David P. Wolfhard, Erol Fikrig

**Affiliations:** 1Section of Infectious Diseases, Department of Internal Medicine, Yale University School of Medicine, New Haven, CT 06519, USA; 2Center for Animal Health Research (CISA-INIA/CSIC), 28130 Madrid, Spain; 3Faculty of Engineering Sciences, Institute of Pharmacy and Molecular Biotechnology, 69120 Heidelberg, Germany

**Keywords:** mosquito saliva, immune responses, arboviruses, vaccines

## Abstract

Arthropod-borne viruses present important public health challenges worldwide. Viruses such as DENV, ZIKV, and WNV are of current concern due to an increasing incidence and an expanding geographic range, generating explosive outbreaks even in non-endemic areas. The clinical signs associated with infection from these arboviruses are often inapparent, mild, or nonspecific, but occasionally develop into serious complications marked by rapid onset, tremors, paralysis, hemorrhagic fever, neurological alterations, or death. They are predominately transmitted to humans through mosquito bite, during which saliva is inoculated into the skin to facilitate blood feeding. A new approach to prevent arboviral diseases has been proposed by the observation that arthropod saliva facilitates transmission of pathogens. Viruses released within mosquito saliva may more easily initiate host invasion by taking advantage of the host’s innate and adaptive immune responses to saliva. This provides a rationale for creating vaccines against mosquito salivary proteins, especially because of the lack of licensed vaccines against most of these viruses. This review aims to provide an overview of the effects on the host immune response by the mosquito salivary proteins and how these phenomena alter the infection outcome for different arboviruses, recent attempts to generate mosquito salivary-based vaccines against flavivirus including DENV, ZIKV, and WNV, and the potential benefits and pitfalls that this strategy involves.

## 1. Emerging Arboviruses, an Increased Concern for Public Health

Newly emerging and re-emerging arboviral infections continue to pose significant public and animal health threats. Changes in global temperature, environmental alterations such as deforestation and fast urban development, and socio-economic conditions have facilitated the rapid circulation of arboviruses within the human populations. All these factors have expanded the geographical distribution and abundance of arthropod vectors, such as some species of *Aedes* (*Aedes aegypti* and *Aedes albopictus*) and *Culex* (*Culex tarsalis*, *Culex quinquefasciatus*, and *Culex pipiens*), which are main transmitters of the most clinically important arboviral diseases [1]. Most of these diseases remain neglected and are impacting both the economy and global healthcare systems, especially in developing countries. Dengue is the most prevalent human arboviral infection transmitted by *Aedes* mosquitoes, striking populations in tropical areas in the Americas, Africa, and South-East Asia, but also generating outbreaks in non-endemic regions, such as those that have occurred in the Mediterranean areas [2]. The past decades have seen a 30-fold increase of dengue cases [3]. More than 3.9 billion people in over 129 countries are at risk of contracting dengue, with an estimated 96 million symptomatic cases and an estimated (World Health Organization and Center for Disease Control and Prevention) 20,000–40,000 deaths every year [4]. Dengue viruses comprise four genetically and serologically related viruses known as serotypes 1 to 4. Infection by any of the four serotypes can result in a range of clinical manifestations, from asymptomatic to self-limiting febrile illness (known as dengue fever), or life-threatening disease characterized by increased vascular permeability, thrombocytopenia, and hemorrhage recognized as severe dengue. It is also known that previous infection with one of these serotypes can enhance the disease severity of subsequent infections with other serotypes, a phenomenon named ADE (antibody-dependent enhancement). The suggested mechanism is through non-neutralizing antibodies pre-generated upon infection with a specific virus serotype that bind to the virus surface of other serotypes and facilitate entry into Fc receptor-bearing cells [5]. Other arboviral infections, such as the emergence and explosive spread of Zika virus infections in Latin America and the Pacific Islands, became a worldwide public health concern due to neurological disorders as Guillain–Barré syndrome and neonatal malformations associated with ZIKV infection. ZIKV infection in humans was first described in Nigeria in 1954 [6]. Few human infections were documented over the next decade until the first reported outbreak in 2007 on the Western Pacific Island of Yap in the Federated States of Micronesia [7]. Since then, large epidemics have been reported in many areas, finally emerging for the first time in the Americas in 2016 [8,9,10]. ZIKV is an example of how an arbovirus disease with an initial low incidence can turn into a global epidemic within a short period of time.

West Nile virus (WNV) initially circulated in an enzootic mosquito–host transmission cycle in Africa, the Middle East, and Europe [11] but swiftly spread throughout the North American continent in the last years, resulting in a high proportion of neurological infections caused by lineage 1 viruses. Since its first appearance in the US in 1999, WNV has emerged as the most common cause of epidemic meningoencephalitis in North America [12]. WNV is easily spread by birds, in which crows, magpies and jays, house sparrows, house finches, and grackles are highly competent reservoirs, transmitted by an unprecedented number of mosquito species [13]. By contrast, susceptible mammals, such as horses and humans, do not develop sufficiently high viremia titers to play a significant role in transmission, becoming dead-end hosts. Despite efforts to monitor and control WNV dissemination, WNV is now endemic in North America, with nearly 7 million cases estimated to have occurred since 1999 [14]. WNV is another example of the potential of arthropod-borne viruses to expand rapidly in non-endemic areas.

The constant threat of arbovirus emergence and re-emergence necessitates a greater fundamental understanding of the biology of these viruses, the interactions between viruses and their vectors, the immune responses generated by the host against the pathogens or the salivary antigens from the vector that can contain or enhance virus transmission to the host, and possible countermeasures that can blunt their impact on public health in order to prevent the emergence of new outbreaks.

## 2. Salivary Proteins Facilitate Mosquito Blood Feeding

To successfully intake blood from a host, the mosquito needs to overcome vasoconstriction, coagulation, and platelet aggregation [15] (Figure 1). Salivary protein secretion is critical for hematophagous insects to assist in blood vessel localization, blood ingestion, and digestion [16]. Several enzyme families exist in mosquito saliva and have been identified among different species [17,18,19,20,21,22]. Among them, apyrase is a soluble secreted protein with ATP-diphosphohydrolase activity, which belongs to the 5′nucleotidase family that hydrolyzes ATP and ADP to AMP and orthophosphate [23]. Apyrase facilitates hematophagy through the inhibition of ADP-mediated platelet aggregation in the host [24]. The expression level of apyrase is induced in the *Ae. aegypti* female salivary gland after blood feeding [25]. Besides *Aedes* mosquitoes, 5′nucleotidase family cDNA was identified in *Anopheles gambiae* [22]. However, 5′nucleotidase family members’ transcripts were not found in *Cx. tarsalis,* and low expression is observed in *Cx. quinquefasciatus* [17]. Lack of this enzyme in this genus may possibly be due to their feeding behavior, as *Culex* mosquitoes are predominately bird feeders and do not need to face the same platelet barrier [26].

Adenosine deaminase and purine hydrolase enzymes have also been identified in *Ae. Aegypti* saliva [19], and their enzymatic activities from *Aedes* salivary homogenate showed the hydrolyzation of inosine and guanosine to hypoxanthine and xanthine [27]. Considering that adenosine and inosine increase cutaneous vasopermeability by mast cell activation [28], the adenosine deaminase and purine hydrolase functions are crucial to decrease itching sensation. Other enzymes, such as the endonucleases, cleave polynucleotides. These have been identified in *Cx. Quinquefasciatus* but not in *Aedes* or *Anopheles*, which may assist blood meal intake by decreasing the blood viscosity [29]. Esterases, another type of hydrolase enzymes, have been shown to be secreted in *Ae. aegypti* saliva. Moreover, the salivary gland lysates from females had higher specific activity than those from males [30], suggesting their potential role in blood feeding.

The vasodilator sialokinin, related to the tachykinin protein family, was found in *Ae. aegypti* salivary glands [31,32]. Sialokinin is crucial during blood feeding by inducing nitric oxide release and stimulating the permeability of endothelial cells [33]. Although *Anopheline* mosquitoes do not produce vasodilatory substances, the salivary peroxidase/catechol oxidase isolated from *An. albimanus* exerts a vasodilatory activity by rescinding hemostatically active biogenic amines that are released during blood feeding [34,35].

Some proteins with protease inhibitor domain have been found in mosquito salivary glands, which might associate with regulating inflammatory processes or host hemostasis. A serine protease inhibitor named anticoagulant factor Xa (AFXa) is expressed only in female salivary glands and inhibits mammalian hemostasis to facilitate blood feeding [36,37]. Another anopheline serine protease is highly conserved across *Anopheles* mosquitoes and inhibits the function of thrombin, causing impaired blood coagulation [38]. Cystatin, with a trypsin inhibitor-like domain and Kazal domain, has been annotated from the mosquito transcripts. The Kazal inhibitor was shown to block the proteolysis function from plasmin recruited by DENV to facilitate infection in the midgut [39]; however, the function of this protein in blood feeding is still unknown.

Another family of abundantly salivary expressed proteins are the D7, pheromone/odorant binding proteins [40]. Two subfamilies of D7 were identified: the short family has a molecular mass of 15–20 kDa and the long is around 27–30 kDa [22,40,41]. The D7 proteins bind to host amines and leukotrienes, antagonizing inflammation and vascular tone [42]. Moreover, a D7 protein from *Cx. quinquefasciatus* was shown to have a high affinity to ADP that inhibits hemostasis and platelet aggregation [43]. The D7 protein acts as an inhibitor of both platelet aggregation and cell recruitment of neutrophils and eosinophils in *Ae. albopictus* [44]. In the *An. stephensi,* the D7 protein blocks bradykinin formation, which is a peptide inducing pain, vascular hypotension, and blood retardation [45]. Other than D7 family, a protein named anophensin has also been identified as kallikrein–kinin system inhibitor resulting in the inhibition of bradykinin release in *An. stephensi* [46].

Aegyptin (belonging to the family of 30-kDa salivary allergens), identified from *Ae. Aegypti*, has also been characterized to prevent collagen-induced platelet aggregation [47,48]. A homolog protein in *An. stephensi* named anopheline antiplatelet protein (AAPP) also has a role as the receptors antagonist that mediates the adhesion of platelets to collagen [49]. Both aegyptin and AAPP have been shown to be essential for maintaining mosquito blood feeding efficiency [50,51].

## 3. Mosquito Salivary Proteins Can Alter the Course of Arbovirus Dissemination and Transmission

Beyond assisting in blood feeding, preliminary work on mosquito saliva or salivary gland extracts show modulatory effects that can influence the infectivity of several mosquito-transmitted viral infections (Figure 2). Intradermal co-injections of Rift Valley fever virus (RVFV) with salivary gland extracts or saliva of *Ae. aegypti* increased mortality rates in mice as well as viremia and virus titers measured in several organs [52]. In the case of dengue, mosquito saliva was shown to enhance infectivity in fibroblasts at early stages of infection [53]. In addition, mosquito bites enhance DENV pathogenesis in humanized mice, maintaining viremia for a longer period and inducing a more severe thrombocytopenia [54]. Similar findings were shown for WNV in C57BL/6 immunocompetent mice bitten by *Culex* mosquitoes, showing an increase in viremia in the early stages of infection [55].

This effect on dissemination enhancement can be triggered in a variety of ways. For example, mosquito bites trigger leukocyte influxes, especially neutrophils, through the activation of CXCL2 and IL-1β expression [56]. Neutrophil recruitment facilitates infection by Semliki Forest virus and Bunyamwera virus to cellular targets including monocytes, acting as trojan horses [56]. Similarly, DENV infection led to the recruitment of inflammatory neutrophils and monocytes to the dermis, enhanced by the presence of *Ae. aegypti* salivary gland extracts in the IFNAR(−/−) mouse model [57]. Subsequent work identified specific salivary proteins which alter this cellular trafficking. For instance, the *Ae. aegypti* salivary protein AgBR1 was found to stimulate the expression of IL-1b and IL-6, produced in response to inflammatory agents, and to induce neutrophil infiltration at the mosquito bite site, significantly enhancing ZIKV pathogenesis in the immunocompromised AG129 mouse model [58]. Another mosquito salivary protein, termed Nest1, was also found to enhance ZIKV disease by inducing the expression of cytokines in Nest1-stimulated neutrophils, such as IL-1b and Cxcl2 and Ccl2 chemokines [59], molecules that may lead to an influx of myeloid cells susceptible to flaviviruses, as mentioned before.

Additionally, mosquito saliva can cause disruption of the endothelial barrier and generates edema. The endothelial barrier of blood vessels acts as a physical defense against virus invasion, separating the skin from systemic blood circulation [60]. Mosquito bites elicit edema that can retain more virus in the skin, helping the infection of cutaneous cells, as was shown for Semliki Forest virus and Bunyamwera virus [56]. Salivary gland extracts (SGE) directly disrupt endothelial barrier function in vitro at the basolateral and apical side of human dermal microvascular endothelial cells and induces endothelial permeability in vivo within the skin microvasculature, increasing the leak of plasma into the dermis. This effect boosts the infection of dermal dendritic cells and macrophages and increases cell migration to skin-draining lymph nodes [57]. Another observation showed mosquito saliva can modulate the permeability of the blood–brain barrier in RVFV-infected mice, significantly increasing viral titers in the brain and triggering its dissemination to the central nervous system [52]. One proposed mechanism that leads to endothelial barrier disruption lies in the serine proteases secreted in the saliva. Serine proteases break down the extracellular protein matrix by endoproteolytic reaction. One of these proteases has been identified in the *Ae. aegpyti* salivary extracts CLIPA3. Mouse embryonic fibroblasts (MEFs) treated with SGE from mosquitoes knocked down for CLIPA3 presented lower DENV infection rates compared with MEFs treated with SGE from wild type mosquitoes [53], suggesting a critical role of CLIPA3 on virus infection to the host. Sialokinin, another *Aedes* gene product, was shown to mediate the enhancement of in vivo virus infection through a rapid reduction in endothelial barrier integrity and by inducing the recruitment of myelomonocytic and myeloid CD11b cells [61].

Innate immune responses represent the first line of defense against pathogens in the host, playing a crucial role in controlling viral infection. Mosquitoes, as other blood-feeding arthropods including ticks or sandflies, also secrete saliva to modify the host immune response, inadvertently suppressing antiviral innate immune responses and modulating early viral replication [62,63,64]. An *Ae. aegypti* 34 kDa protein was shown to increase DENV replication in vitro in human keratinocytes, the most common type of skin cell, inhibiting the transcript expression of different antiviral molecules, such as IFN-α, IFN-β, IRF3, IRF7, LL-37, and S100A7 [65]. The salivary factor LTRIN interferes with the lymphotoxin-β receptor pathway, inhibiting its dimerization in the presence of its natural ligand, lymphotoxin-β, and attenuating the activation of NF-κB and the production of proinflammatory cytokines. The final consequence is the enhancement of ZIKV infectivity in both in vitro (THP-1 and HUVEC cells, fibroblasts, and bone marrow-derived macrophages) and in vivo (IFNAR(−/−) mouse) models, in which higher viral titers and severe body weight loss were observed after LTRIN inoculation [66]. *Aedes* AaVA-1 protein activates autophagy to promote flaviviral infection. It induces the conversion of LC3B-I, central protein in the autophagy pathway, into LC3B-II in THP-1 cells, and facilitates in vitro infection of ZIKV, DENV, and Semliki Forest virus. AaVA-1 also potentiates ZIKV infection in mice, showing higher viremia levels and mortality in ZIKV-infected animals co-inoculated with AaVA-1 [67]. An interesting fact that has been poorly studied is how these salivary proteins can be internalized into the cells by endocytosis. In this work, authors also showed how AaVA-1 gains access into human monocyte-derived macrophages by a RhoA-dependent endocytosis and colocalized intracellularly with the early/late endosomes and Tom20, a mitochondrial marker. These findings suggest that the intracellular trafficking route of AaVA-1 is to escape from endosome to mitochondria, as was observed for some external toxin proteins, such as the Shiga toxin B-fragment [68].

Adaptive immune responses are also critical in controlling virus infection in the host. *Aedes* saliva has immunomodulatory effects on dendritic cells, crucial for priming adaptive T and B cell responses, triggering the migration of activated dendritic cells from the skin to the draining lymph node [57], and therefore facilitating virus dissemination. Mosquito SGE also suppresses T cell proliferation in murine splenocytes. In addition, these cells reduced the production of both interferon-g (IFN-g) and interleukin-10 (IL-10) when cells were pre-exposed to *Aedes* SGE and induced cell death in CD4+ and CD8+ T cell populations [69]. Although this phenotype was not observed when SGE from *Culex* mosquito was used [69], reduced numbers of T cells were counted in skin tissue explants in mice intradermally infected with WNV and bitten by *Culex* mosquitoes [70], observing higher WNV replication in the skin and in the lymph nodes.

Surprisingly, some mosquito salivary proteins may inhibit disease progression. D7 proteins, which have been described as biogenic amines and cysteinyl leukotrienes binders, can inhibit virus infection in vitro and in vivo. Human monocytic U937 cells infected with DENV and either pre-treated or co-treated with recombinant D7 protein from *Aedes aegypti* resulted in lower DENV2 titers. Similar observations were found in vivo, where AGB6 mice co-inoculated with D7 and DENV2 presented lower titers in footpads and draining lymph nodes at 48 h post-infection (hpi) [71]. In addition, research performed using D7 proteins from *Cx. quinquesfaciatus* suggested a similar effect in the viral outcome produced by WNV in the mouse model, as anti-D7 antibodies passively transferred to mice enhanced the disease transmitted by infected mosquitoes [72]. Another *Aedes* protein, aegyptin, was shown to block the interaction of collagen with its physiological ligand, subsequently interfering with platelet aggregation and adhesion [47,48]. Despite this role, aegyptin resulted in decreased DENV infection in vivo at the inoculation sites and blood at 48 hpi [73], suggesting an intricate relationship between mosquito salivary protein and the virus as well as the host.

Collectively, there is a complex interplay between components of mosquito saliva and host factors that can influence the success of virus transmission to humans during mosquito blood feeding. Taking steps to elucidate the role of mosquito saliva components in modulating human hemostasis and immune responses is imperative for understanding how the virus hijacks this mechanism to facilitate the success of viral infection in humans. Armed with this knowledge, and always considering the limitations of the work with immunocompromised animal models, we could then improve the development of blocking therapies such as vector-based vaccines or drugs against mosquito-borne viral diseases.

## 4. The Use of Salivary Proteins as Vector-Based Vaccines: Benefits and Potential Pitfalls

Vaccines are available for only three mosquito-borne flavivirus infection (Figure 3). First, the live-attenuated 17D vaccine has been used since the 1930s against YFV [74]. Second, several vaccines are available for JEV which can be broadly grouped into three types—(1) inactivated JE vaccine such as MB-JEV, ENCEVAC, and IXIARO, (2) live-attenuated JE vaccine SA-14-14-2, and (3) live YFV-JEV chimeric vaccine. Thorough literature summaries on JEV vaccines are available [75,76] and this topic will not be covered in detail in this review. Finally, several vaccines for DENV raise worldwide attention. Dengvaxia, a live attenuated chimeric vaccine that is designed to work against all four DENV serotypes, was approved for use in around 20 countries up to 2019 [77]. Additionally, DENVax recently received approval for use in Europe and Indonesia in 2022 [78,79]. Albeit rare, 17D is associated with neurotropic and viscerotropic diseases occurring at an average of five cases per million doses [80], with <5% fatality rate in a case study [81]. In addition, controversy was raised against Dengvaxia as vaccinees with no prior DENV exposure developed severe dengue when infected with the virus [77]. The United States FDA and the WHO revised the guidelines for Dengvaxia to be administered only to those who have been exposed to DENV infection either by health history or serological assessment [82,83]. These cases underscore the need for a better vaccine design that is safe for administration in humans. Additionally, ongoing threat still persists for the transmission of other flaviviruses such as WNV and ZIKV, despite a decrease in reported cases in the US since 2017 (ZIKV) and 2018 (WNV) [84], especially since severe cases of flavivirus infection can lead to fatal neurological disease in the case of WNV or severe neurological symptoms such as Guillain–Barré syndrome, neuropathy, and myelitis in the case of ZIKV. Taken together, there is an urgent need for vaccine development to control the spread of mosquito-borne flavivirus infection, especially in endemic area of the world.

The traditional strategy for vaccine development against flavivirus transmission is by using antigenic viral component as the core of the vaccine, and this strategy is still being used today (reviewed in [85,86,87]). Evidence in recent years unveiled that this strategy is hindered by rapid viral genome mutation and the subsequent emergence of new viral subtypes, which lead to the mechanism of both immune and vaccine escape for vaccines still under development (reviewed in [88]). Furthermore, the occurrence of antibody-dependent enhancement provides more evidence of the many mechanisms utilized by the virus to escape clearance by the host immune cells. This scenario is observed in heterotypic secondary DENV infection [89], and subsequent ZIKV infection after first WNV infection [90,91] or DENV infection [91]. Therefore, a novel approach to vaccine design is necessary. The idea of using a vector-based vaccine becomes popular with the underlying premise that it shifts evolutionary pressure away from the virus (Figure 3). It is unlikely for the vector, in this case the mosquito, to generate a mutation as it does not give an evolutionary advantage and is also not feasible due to longer life cycle of the mosquito. Indeed, the first proof of principle on mosquito saliva-based flavivirus vaccine was performed in 2013 when immunization with mosquito salivary gland extract was shown to be able to prevent WNV infection in Swiss mice [92].

Moreover, vector-based vaccines can be intended to target multiple flaviviruses. This is because several flaviviruses are carried by a common mosquito vector. As an example, ZIKV and DENV are carried by *Ae. aegypti* mosquito. It is then hypothesized that proteins of mosquito saliva from *Ae. aegypti* can be used to develop a vaccine that can control the transmission of both DENV and ZIKV. Preliminary evidence for the potential of using mosquito saliva-based antigen as pan-flavivirus vaccines in multiple studies have been shown. First, passive immunization with NeSt1 and AgBR1, both secreted proteins in mosquito saliva, individually or in combination [58,59,93], can reduce ZIKV infection in Ag129 mice. In addition, vaccination with recombinant AgBR1 can decrease ZIKV infection in Ag129 mice [94]. Interestingly, passive immunization with AgBR1 antiserum can also reduce WNV infection by *Ae. aegypti*-borne infection in a lab-controlled environment [95], providing preliminary evidence that a mosquito saliva-based vaccine can be used to target multiple flaviviruses. However, more studies need to be conducted to see whether AgBR1 and NeSt1 can prevent DENV infection in mice. Nevertheless, these studies show promising views on the potential of a mosquito saliva-based vaccine to be used as a pan-flavivirus vaccine.

Vector-based vaccines can also modulate virus transmission by deterring the virus from completing its life cycle in the vector, a type of vaccine termed transmission-blocking vaccine (TBV) [96,97]. This vaccine type is especially useful in preventing further spread of the virus in an already endemic population and may not necessarily require human vaccination. Instead, wild animals or livestock can be the target of this vaccine. An example of TBV was shown when antiserum against cysteine-rich venom protein CRVP379, which was shown to be expressed in both the salivary gland and midgut, introduced in Aag-2 cells or *Aedes* mosquito inhibited DENV infection [98]. Another salivary protein, AaSG34, was shown to cause reduction of DENV infection in *Aedes* mosquito [99] and human keratinocytes [65] upon silencing. These suggest a dual role of AaSG34 in modulating viral infection in both host and vector, underscoring the potential use of this salivary antigen for salivary-based vaccine candidates.

While there are many factors that favor the shift towards the development of vector-based vaccine as discussed above, there are also other factors that necessitate careful assessment to further refine the feasibility of such vaccine approaches. First and foremost, there is not enough evidence on whether a vector-based vaccine is safe for human administration, as most research for vaccines based on mosquito saliva has been conducted in murine models. In the first of its kind, the collaborative group from NIAID and London researchers published a phase I clinical trial on *An. gambiae* saliva vaccine (AGS-v) in 2020 aimed at testing the tolerance and immunogenicity of AGS-v in human [100]. AGS-v consists of four salivary peptides derived from *An. gambiae* salivary glands but are common amongst *Anopheles* spp., *Aedes* spp. and *Culex* spp. In this randomized and double-blind study, AGS-v was tested either alone or with Montanide ISA 51 as an adjuvant with placebo as a control group. They found that AGS-v alone is well tolerated and subsequent *Ae. aegypti* mosquito bite did not induce severe reaction around the bite site among study participants. Additionally, adjuvanted AGS-v is highly immunogenic and can induce IFN-gamma response among study participants. Recently, the same group performed the same study on the same vaccine with an added fifth peptide, AGS-v PLUS, and reached the same conclusion [101]. In addition, the authors showed in vitro evidence that AGS-v PLUS can prevent ZIKV infection after treating Vero cells with peripheral blood mononuclear cells (PBMCs) and serum from immunized individuals [101]. While both studies were conducted with a limited number of study participant and have not shown in vivo evidence for protection against different flaviviruses in human, they nonetheless provide a glimpse of a promising view for using mosquito saliva as a vaccine for flavivirus diseases. To date, there are no other clinical trials on vaccines based on mosquito saliva. Indeed, more in-depth study is needed, ideally with a larger cohort and diverse patient demographic that includes children and pregnant women.

In addition, it is not clear how to best select the mosquito saliva protein to be used in vaccine design. Vaccine efficacy against a specific arbovirus may change against others, as not all arboviruses interact with the host immune system in the same manner. While we have shown evidence that several mosquito saliva proteins could potentially be used in a vaccine, there are other candidates that do not show any effect despite being highly immunogenic ([58], data not shown). One particular study showed that using mosquito saliva protein as a vaccine can exacerbate disease progression in mice [72]. In the study, as we mentioned in the previous section, D7 salivary protein of *Cx. tarsalis* was selected by the authors on the basis of abundance in mosquito saliva and highly immunogenic effect upon mosquito bite, which is then tested for its ability to prevent WNV infection in mice. Both passive immunization with recombinant D7 (rD7) antiserum and active immunization with rD7 showed an increase in mortality upon mosquito bite-delivered WNV infection in mice. Interestingly, mice immunized with rD7 showed an increase in IgG1 production and inflammatory cell infiltration at mosquito bite site, creating a favorable condition for WNV infection. This suggests a complex interplay between the mosquito salivary protein and its immunomodulatory effect in the host during mosquito blood feeding and how it subsequently affects virus dissemination. Therefore, careful selection of salivary protein to be used in vaccine design is imperative during the making of vector-based vaccines.

Finally, mosquito saliva-based vaccines can prevent the infection of flavivirus that occurs through mosquito bite but not by needle injection [58]. It is therefore unlikely that this type of vaccine can prevent flaviviral infection that is transmitted by means other than mosquito bite. In the case of ZIKV, transmission of ZIKV can also occur through sexual intercourse, blood transfusion, and from mother to child during pregnancy or childbirth. WNV can be transmitted by blood transfusion and from mother to baby during pregnancy, delivery, or breast feeding, as well as DENV [102]. It is unlikely that a vector-based vaccine can prevent the spread of ZIKV through these modes of transmission. However, vector-based transmission still accounts for the majority of flaviviral diseases worldwide [102]. This observation does not preclude vaccine development that is based on mosquito saliva. Nonetheless, future recipients of vector-based vaccines should be made aware of the pitfalls of this type of vaccine.

## 5. Immune Responses to Saliva Proteins as Biomarkers of Exposure

*Aedes* mosquitoes cause major health problems through the arboviruses that they transmit, especially but not exclusively in developing countries. Inoculation of mosquito salivary protein into the host at the bite site by hematophagous mosquito induced immune response followed by generation of antibodies. Indeed, several studies found some mosquito salivary proteins to be highly immunogenic among human populations living in geographical areas teeming with mosquitoes [103,104]. Hence, the use of IgG antibodies against specific *Ae. aegypti* saliva proteins could prove to be reliable indicators for the detection of exposure to *Aedes* bites, to assess the spread of mosquitoes and the success of mosquito population control [105], and as biomarkers to monitor disease severity [106]. A recent study showed that high levels of antibodies to *Ae. aegypti* salivary proteins are associated with the future development of dengue infection [107]. The role of several salivary proteins has been evaluated as a biomarker of mosquito exposure and arbovirus disease. For instance, the 34k2 salivary proteins from *Ae. albopictus* and *Ae. aegypti* have been described as suitable candidates for the development of serological assays to evaluate spatial and/or temporal variation of human exposure to *Aedes* vectors [108,109,110]. The use of antibodies against the Nterm-34 kDa peptide has been postulated to distinguish dengue disease progression, although no conclusive connection to dengue risk has been shown [111]. Nevertheless, specific antibodies against D7L positively correlate with DENV-positive and febrile patients as compared to non-febrile subjects [112], whereas antibodies against AgBR1 salivary protein were significantly different in dengue severity classification. Patients with dengue without warnings present higher anti-AgBR1 IgG levels than patients with dengue with warnings, indicating that the higher the antibodies, the less severe the symptoms/infection. This analysis suggests that AgBR1 IgGs may be a diagnostic tool to evaluate the risk of dengue fever severity in endemic regions [111].

Similarly, in the case of malaria, human humoral immune responses in the form of specific IgG antibodies against *An. gambiae* whole saliva [113,114] and several saliva proteins from *An. gambiae* have been considered as promising biomarkers for risk assessment of malaria transmission and severity in Senegal and the Brazilian Amazon [85,86]. To date, some studies have measured anopheline biting rates and the prevalence of human antibodies to the salivary antigen gSG6 [115,116]. This indicator could represent an alternative to classical entomological and parasitological monitoring methods for measuring and following the effectiveness of vector control strategies. This salivary protein was also tested as a potential biomarker of exposure for other *Anopheles* species [117]. However, there is a lack of association between anti-*An. gambiae*-gSG6 antibody titers and concurrently measured human biting rates for *An. farauti* exposure, suggesting that the assay for human anti-*An. gambiae*-gSG6 antibodies lacks sufficient sensitivity to be a multi-anopheline biomarker. These findings imply that an improvement in the sensitivity of serology to monitor changes in anopheline biting exposure may require the use of saliva antigens from local anopheles species, and this may be especially true for species more distantly related to the African malaria vector *An. gambiae* [115].

Additionally, proteomic approaches have been applied to discover new putative biomarkers of risk of malaria infection in the saliva of *An. albimanus* that were immunogenic in humans. Antibody levels against salivary proteins PEROX-P3, TRANS-P1, and TRANS-P2 were significantly higher in serum samples from malaria-infected individuals compared to samples from uninfected individuals. Therefore, the use of these proteins as biomarkers of both exposure to New World *Anopheles* bites and malaria transmission risk could serve as important tools in malaria surveillance and control programs [118].

## 6. Conclusions and Future Challenges

The emergence and re-emergence of mosquito-borne diseases, especially during the last decades, have necessitated rapid development of new preventive strategies to control the transmission of these pathogens within human populations. It is evident that vector saliva components influence successful early viral infection and dissemination in the host. However, the functions of many salivary proteins remain unclear. There are at least 1000 salivary proteins discovered using high-throughput proteomic and transcriptomic approaches in *Aedes aegypti* saliva. The abundance of proteins present in mosquito saliva hampered the mechanistic study of how these proteins modulate host hemostatic mechanisms and thereby influence the outcome of the disease. Having this information is imperative as it provides foundational knowledge for the development of mosquito salivary-based vaccines. Although there are still obstacles to overcome, the current state for salivary vaccine development is exciting as experiments with animal models show promising results and human clinical trials show the safety and immunogenicity of these vaccines. Additionally, these vaccines could also be complemented with viral antigens to augment their efficacy against one specific virus. Finally, humoral immune responses against mosquito salivary antigens can be used as a biomarker of exposure to the vectors and of disease risk, which is especially useful for developing and endemic countries where resources are limited.

## Figures and Tables

**Figure 1 pathogens-12-00371-f001:**
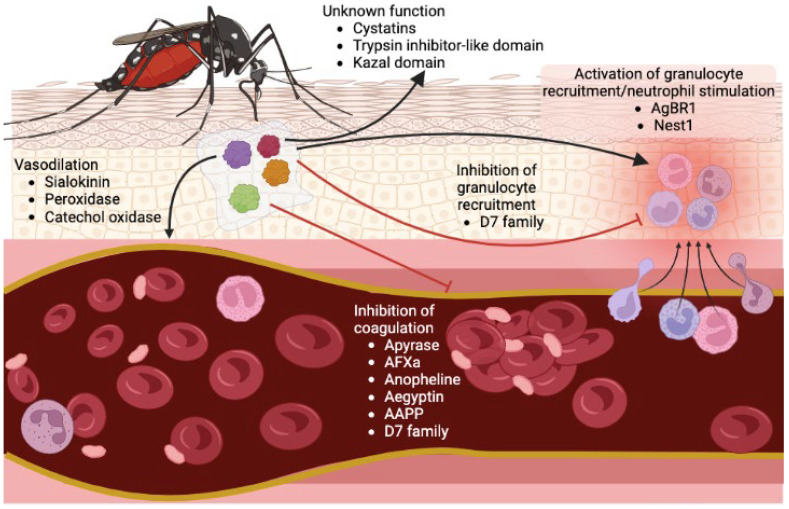
Schematic representation of the physiological effect performed by the mosquito bite on the host skin. The image shows the effect of some of the salivary proteins of the mosquito and their role in blood intake.

**Figure 2 pathogens-12-00371-f002:**
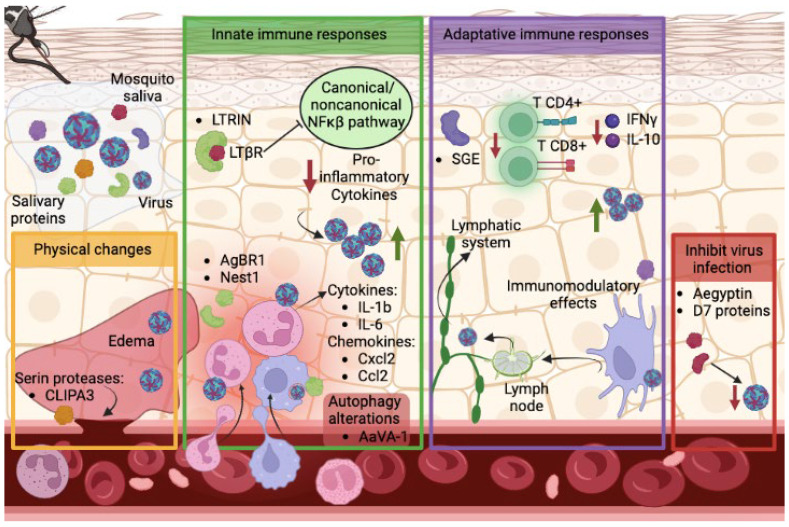
Schematic representation of the interactions between mosquito salivary proteins, flavivirus, and the host immune and homeostatic responses. The effect of these mosquito salivary proteins in many cases facilitates infection of the host by altering both its innate and adaptive immune responses. On the contrary, also during a natural infection, other salivary proteins are able to block the dissemination of the virus.

**Figure 3 pathogens-12-00371-f003:**
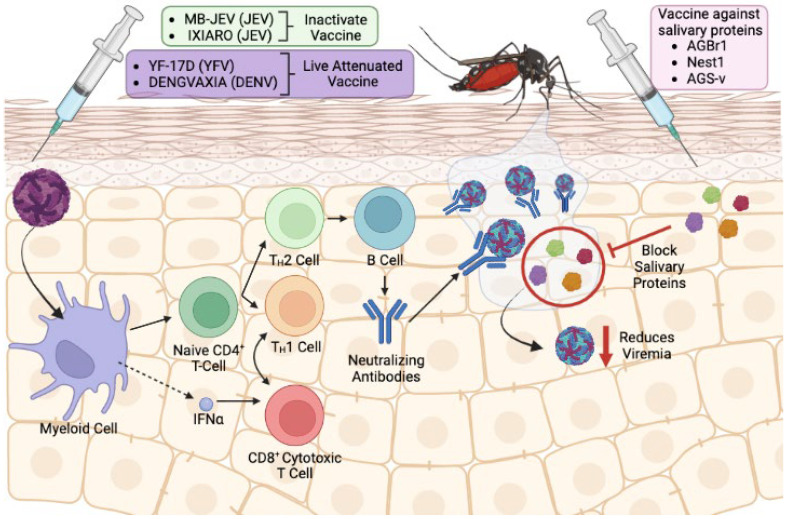
Schematic representation of the different existing vaccination mechanisms against viruses transmitted by arthropods. The left side of the image portrays the mechanisms on which classical commercial vaccines are based. These are live attenuated or inactivated vaccines whose target is the virus component itself. On the right side, new generation vaccines based on the salivary proteins of the mosquito and whose target is to generate a response against the salivary proteins and thus prevent the replication of the virus.

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
