# Peer review of "Mosquito Salivary Proteins and Arbovirus Infection: From Viral Enhancers to Potential Targets for Vaccines"

_pathogens, 2023, doi:10.3390/pathogens12030371_

Round 1

Reviewer 1 Report

This review article discusses mosquito salivary proteins and their ability to either enhance or protect against infection by arthropod-borne viruses.

While the review is interesting, the English needs to be checked for spelling (e.g., human when it should be humans), and words missing (e.g. antibody dependent… but no “enhancement” after the first two words).

I think the authors need to decide the focus of the paper. The title says “flaviviruses” but an alphavirus (Semliki Forest) and two bunyaviruses are included. The term “arboviruses” is used a lot, which makes it confusing. I suggest either the authors either focus on flaviviruses or expand to include other mosquito-borne viruses.

The examples provided are interesting but I think the authors should be precise on the mouse model for each example as interpreting innate responses in immunocompromised mouse models for some viruses is not straightforward as written. Some of the text related to conclusions for mouse models needs to be toned down where immunocompromised mice were used.

A number of the references are incomplete. Please check the reference list.

The Introduction section on pages 1-2 is a bit sensationalism (e.g., “devastating”, “striking”), which could be toned down.  

Page 1, 6 lines from the bottom: the term “developing countries” is not used these days. “Low income countries” is preferred.

Page 2, line 1: 40,000 deaths is too high. Most believe it is less than 20,000.

Page 2, paragraph 2, line 3: “last decades”. This is a bit misleading since the virus arrived in 1999.

Page 4, paragraph 1: I would suggest this paragraph is revised to include whether or not there is any evidence from humans that there is enhanced disease or is it just seen in mouse models.

Page 7: the vaccine section needs some revisions. 17D was not derived from live attenuated virus. “several” vaccines are available for JEV but this needs to be explicitly stated… live SA14-14-2 and inactivated cell culture vaccine. Note Figure 3 has “MB-JEV”. I assume this means mouse brain derived inactivated vaccine. This is only used in a couple of countries today; I would revise. “<5%” rate for SAEs for YF 17D vaccine seems very high to me as that equates to 1 in 20 vaccinees, which is not correct. The sentence on dengvaxia is wrong. The vaccine did not cause the dengue disease, rather  a few DEN seronegative vaccinees got dengue disease 2 years after their third dose of vaccine. The Takeda DENVAX Live attenuated vaccine has been approved in the EU and Indonesia in 2022 and is probably worth a mention. Paragraph 2, line 1 states vaccine development was…. “initially focused”. I would say it still is. Please revise. The sentence on lines 2-4 on paragraph 2 is incorrect. There is no evidence of any genetic variation in a flavivirus resulting in a licensed vaccine not working, except may be JEV genotype V but that is a discussion point for those who work on JEV.   

Reviewer 2 Report

The review article "Mosquito salivary proteins and flaviviral infection: from viral enhancers to potential targets for vaccines." by Alejandro Marin Lopez * , Hamidah Raduwan , Tse-Yu Chen , Sergio Utrilla Trigo , David Phillip Wolfhard , Erol Fikrig * provide sveral information concerning the mosquito viral proteins as a candidate vaccine in future. The comment for authors as followed: 1. There is no new information about the prospective candidate vaccine of salivary protein. 2. The controversial research of salivary protein as candidate vaccine was still crucial for different viruses in Flaviviruses. 3. The promising of using salivary protein as a biomarker to monitor mosquito bite in humans should be possible to explore. 4. There is no more data to add on the knowledge of salivary protein vaccine in the article. 5. They require more research to give new information for the future salivary protein vaccine.    

Reviewer 3 Report

The authors conducted a thorough and well-organized review of the functions of mosquito saliva in transmission of arboviruses and they detailled how this knowledge can be harnessed as a vaccine strategy. The paper is very well written and I have just minor editorial points.

Minor comments :

When talking about a trypsin inhibitor-like domain and Kazal domain, the authors can discuss the observation that a Kazal-type proteins from the mosquito midgut (that is the same as in saliva) binds DENV to facilitate tissue penetration in midgut (Ramesh et al; 2019. iScience). This could provide cues about the function in saliva.

 “Another protein named anophensin, although is not belong to the D7 family,” I think there is a grammar mistake

 “the occurrence of antibody-dependent provides more evidence of the many mechanisms utilized”; a word is missing.

 “the use these proteins as biomarkers of both exposure” a word is missing

Round 2

Reviewer 2 Report

Minor notice:

1.  I can spot some mistake or error typing about the word DENVax that I do not know it is the same as Dengvaxia.

2, I found the references 64 and 100 are the same article. Could you please check again for some duplication of other references.

Author Response

We would like to thank reviewers for the minor comments.

1)Dengvaxia is a vaccine candidate based on the backbone of yellow ferev YF17D vaccine, encoding DENV prM and E proteins. In contrast, DENVax, another dengue vaccine candidate, is based on the backbone of the DENV2 PDK-53 attenuated vaccine, encoding prM and E proteins for the rest of serotypes.

2) We have fixed that reference duplication and another one detected in the bibliography.